# The Impact of Ambient Environmental and Occupational Pollution on Respiratory Diseases

**DOI:** 10.3390/ijerph19052788

**Published:** 2022-02-27

**Authors:** Chinatsu Nishida, Kazuhiro Yatera

**Affiliations:** Department of Respiratory Medicine, University of Occupational and Environmental Health, Fukuoka 807-8555, Japan; c-nishi@med.uoeh-u.ac.jp

**Keywords:** ambient pollution, occupational pollution, respiratory disease, lung disease

## Abstract

Ambient pollutants and occupational pollutants may cause and exacerbate various lung and respiratory diseases. This review describes lung and respiratory diseases in relation to ambient pollutants, particularly particulate matter (PM_2.5_), and occupational air pollutants, excluding communicable diseases and indoor pollutants, including tobacco smoke exposure. PM_2.5_ produced by combustion is an important ambient pollutant. PM_2.5_ can cause asthma attacks and exacerbations of chronic obstructive pulmonary disease in the short term. Further, it not only carries a risk of lung cancer and death, but also hinders the development of lung function in children in the long term. It has recently been suggested that air pollution, such as PM_2.5_, is a risk factor for severe coronavirus disease (COVID-19). Asbestos, which causes asbestosis, lung cancer, and malignant mesothelioma, and crystalline silica, which cause silicosis, are well-known traditional occupational pollutants leading to pneumoconiosis. While work-related asthma (WRA) is the most common occupational lung disease in recent years, many different agents cause WRA, including natural and synthetic chemicals and irritant gases. Primary preventive interventions that increase awareness of pollutants and reduce the development and exacerbation of diseases caused by air pollutants are paramount to addressing ambient and occupational pollution.

## 1. Introduction

Air and occupational pollution are well known to negatively affect human health and cause chronic inflammation [1], leading to the development and exacerbation of respiratory diseases. Various human health issues have recently been raised due to urbanization, industrialization, population growth, climate change, and aging of the working population [2,3]. According to the World Health Organization, air pollution accounts for >6.5 million deaths worldwide per annum, equating to 11.6% of all global annual deaths [2]. Air pollution, particularly particulate matter_2.5_ (PM_2.5_) (≤2.5 µm in aerodynamic diameter), is a leading cause of several adverse health effects, including pulmonary diseases, and leads to 3.3 million premature deaths per annum worldwide [3]. Occupational air pollution is another crucial public health issue. Exposure to respirable dust, fumes, vapors, gases, volatile organic compounds, chemicals, and metals is hazardous and may cause occupational lung diseases [4]. Occupational lung diseases include diverse diseases such as asthma, chronic obstructive disease (COPD), pneumoconiosis, hypersensitivity pneumonitis (HP), and malignant diseases (lung cancer and malignant mesothelioma), with work-related asthma (WRA) being the most common [5]. Air pollution and occupational air pollution can be avoided, and primary prevention is critical for their control and prevention.

This review describes lung and respiratory diseases in relation to ambient pollutants, particularly PM_2.5_, and occupational air pollutants, excluding communicable diseases and indoor pollutants, including tobacco smoke exposure. The aim of this review was to understand the current knowledge of environmental and occupational pollution and their adverse health effects based on recent information, as well as to highlight their effects on respiratory medicine and occupational health and increase awareness of active preventive interventions.

## 2. Methods

A search of the peer-reviewed English literature was conducted using MEDLINE (PubMed). The search was limited to reviews, systematic reviews, meta-analyses, randomized controlled trials, clinical trial, and books and documents published between 1 January, 2017, and 1 December, 2021, using the following search terms: (environmental) AND (occupation) AND (pollution) AND (lung OR respiratory OR pulmonary). A total of 275 extracted studies, 20 studies were screened by title and abstract, then 13 studies were extracted by full text, and eventually 8 studies were included in this review.

## 3. Ambient Pollution and Occupational Pollution

### 3.1. Ambient Pollution

Environmental pollution, including air pollution (ambient indoor and outdoor pollution), water pollution, and soil pollution, is well known to cause various health effects in humans. Air pollution can damage almost any organ system and cause acute or chronic lung and heart diseases [1,6]. Air pollutants are defined as any ambient substances that may be harmful to humans, animals, plants, or materials [6]. Such pollutants emerge from various sources, including industry, transportation, soil, volcanoes, and the ocean, each of which has different characteristics depending on its components, such as biological materials and aeroallergens [1,6]. In addition, air pollution is a complex mixture of different gaseous and particulate components, the composition of which varies widely by region and season [7]. However, it is difficult to unequivocally distinguish between particulate and gaseous matter, such as the transition from gaseous materials to particles or from particles to gaseous materials, depending on atmospheric conditions [8]. Typical air pollutants are composed of particulate matter such as PM_2.5_ or PM_10_; gases such as SO_2_, NO_2_, O_3_, CO, reactive hydrocarbons, and polycyclic aromatic hydrocarbons (PAHs) such as naphthalene or anthracene; and volatile organic compounds (VOCs) such as toluene, xylene, and ethyl acetate, which are generally produced by industrial or transportation sources [1,6,9]. Over the last few decades, while emissions and air concentrations of traditional industrial air pollutants such as SO_2_ and smoke particles have declined in developed regions, problems with the emissions of vehicle engine combustion products such as VOCs, NO, and fine particulates from vehicle exhaust and secondary ozone are persisting and even increasing [7]. 

PM_2.5_ is considered to be the most harmful PM and is subject to several air pollution regulations [1,10]. PM_2.5_ is a PM with an aerodynamic diameter of <2.5 μm [11] and includes several pollutants: heavy metals, PAHs, unburned or incompletely burned fuel gases, sulfur compounds, and nitric compounds [12]. PM_2.5_ includes PMs that are directly produced by combustion (so-called ‘primary particulates’) and those that are produced, largely from primary pollutants, by chemical reactions in the atmosphere (known as ‘secondary particulates’). Sources of primary particulates include facilities that generate soot; smoke induced by boilers and incinerators; and dust (fine dust) and mineral deposits created by coke ovens, automobiles, ships, aircraft, soil, and oceans. There are also natural gaseous and dust materials created by volcanoes and their cross-border pollution. Secondary particulates are produced from sulfur oxides (SO_x_) and nitrogen oxides (NO_x_) emitted by fuel combustion in thermal power plants, factories, business establishments, automobiles, ships, aircraft, and households, and from gaseous substances, such as VOCs emitted from evaporation from solvents, petroleum handling facilities, forests, and fuel combustion facilities, by reacting with light and ozone in the atmosphere [8]. Gaseous and particulate materials are also produced from smoking, cooking, and stoves in indoor environments.

PM_2.5_ can deeply penetrate the lungs with a size threshold of deposition in the alveoli [1,10], which may initiate harmful airway effects, including airway oxidative stress, lung and systemic inflammation, cilia dysfunction that increases the risk of infectious diseases, and increased airway hyperresponsiveness [10]. The harmful effect of PM generally depends not only on the particle size, but also on its structure and composition. There may also be toxic components on the surface of the particles, which can induce tissue damage. Toxic elements (e.g., arsenic, lead, or cadmium) or compounds (e.g., sulfuric acid or polycyclic aromatic hydrocarbons) can adhere to the surface of the PM during the combustion process and be carried deep into the lung; ultrafine particles are of particular concern. This situation is most closely related to the particles that emerge from coal combustion, which contain many heavy metal components and high levels of sulfur compounds. In addition, air pollution, including PM, may interact with airborne allergens to induce and exacerbate allergic asthma [13]. 

The biological effects of PM_2.5_ have generally been reported as short-term or long-term exposure effects (Table 1). Short-term (hours to days) exposures have been reported to increase hospital and emergency visits due to asthma attacks or COPD in many meta-analyses [14,15,16,17]. In asthma, with an increase in the PM_2.5_ concentration of 10 μg/m^3^, the risk of hospital visits has been shown to increase by 1.5% [14]; Zheng et al. [15] and Lim et al. [16] also indicated a relative risk of hospital or emergency consultation of 1.023 and 1.048, respectively. Similarly, in COPD, a 10 μg/m^3^ increase in the PM_2.5_ concentration resulted in a significant increase in the rate of hospital visits (0.31%) [16], and also significantly increased the relative risk of acute exacerbations of COPD (1.031) [17]. In addition, long-term (months to years) exposure to PM_2.5_ has been shown to affect lung function development in children. Over an 8-year period (from 10 to 18 years old), the decreased gain in forced expiratory volume in one second (FEV_1.0_) was significantly associated with exposure to PM_2.5_ (*p* = 0.04), and the estimated proportion of 18-year-old subjects with a low FEV_1.0_ (defined as <80% of the predicted FEV_1.0_) was approximately five-fold greater in those exposed to the highest level of PM_2.5_ than in those exposed to the lowest level (*p* = 0.002) [18]. 

PM was classified as a group 1 carcinogen by the IARC [19], and it has been shown that long-term exposure to PM_2.5_ is associated with the development and mortality of lung cancer. In a large cohort study, the mortality risk ratio of lung cancer increased significantly to 8% [20] and by 1.23-fold [21] when the average concentration of PM_2.5_ increased by 10 μg/m^3^. In addition, a global meta-analysis revealed that PM_2.5_ significantly increased the risk of lung cancer development [22]. 

Each country makes efforts to control the PM_2.5_ level via emission regulation. However, the level may also be affected by climate change and drastic fluctuations during a day or season [23]; thus, it may be difficult to exert control via regulation alone. Greater attention should be paid to the PM_2.5_ level, particularly regarding the populations susceptible to air pollution, for example, children, elderly people, pregnant women, those with cardiovascular or respiratory diseases or diabetes [24,25,26]. 

**Table 1 ijerph-19-02788-t001:** Health hazards to the respiratory system of PM_2.5_.

Adverse Health Effects		Reference Number
Short-term exposure (hours to days)		
Asthma attacks	Increased risk of hospital visits	[14,15,16]
COPD	Increased risk of hospital visits,	[16]
	Increased the relative risk of acute exacerbations	[17]
Long-term exposure (month to years)		
Lung function development	Suppression of lung function development in children	[18]
Lung cancer	Increased the development and mortality of lung cancer	[20,21,22]
SARS-CoV-2 associated events	Increased the risk of SARS-CoV-2-associated respiratory distress, respiratory failure, and mortality	[27,28,29,30,31,32]

SARS-CoV-2: severe acute respiratory syndrome coronavirus 2.

Recently, chronic exposure to air pollution has intensified severe acute respiratory syndrome coronavirus 2 (SARS-CoV-2)-associated respiratory distress, respiratory failure, and mortality [27,28,29,30,31,32]. Chronic exposure to air pollutants such as PM_2.5_ and NO may promote SARS-CoV-2 entry and lifecycle in the body via several mechanisms, including the prevention of antiviral activity by antimicrobial peptides and surfactant protein D, increasing angiotensin-converting enzyme 2 receptor expression levels on the surface of airway epithelial cells, and inhibiting mucociliary clearance [27,33]. Severe acute respiratory illness due to SARS-CoV-2 can worsen Th1/Th17 proinflammatory cytokine production in humans, resulting in a cytokine storm. Air pollutants, such as PM_2.5_, can also enhance the Th1/Th17 immune response and may exacerbate SARS-CoV-2-related respiratory distress, failure, and mortality [34].

### 3.2. Ambient Occupational Pollution

Occupational air pollution may cause or worsen various lung diseases, including malignant diseases such as lung cancer and malignant mesothelioma, WRA, occupational COPD, pneumoconiosis, and HP (Table 2 and Table 3). It is estimated that 5–10% of global cases of lung cancer are due to occupational exposure to carcinogens [35]. Lung cancer is frequently associated with exposure to carcinogenic materials in the workplace [36,37]. Many occupational pollutants cause lung cancer in humans, including heavy metals such as arsenic, chromium [Ⅵ], and nickel and radiation [38], but the most common is asbestos. Asbestos comprises natural silicate mineral fibers that can be categorized into two groups: the serpentine group (e.g., chrysotile) and the amphibole group (e.g., crocidolite, amosite, tremolite, and others). The serpentine group has a winding standard structure, whereas the amphibole group is composed of straight, rod-like fibers, and both forms are related to the development of lung cancer and malignant mesothelioma [39]. These fibers are used in construction and manufacturing, such as brake linings, pads, and insulation. By the mid-20th century, asbestos exposure had drastically increased the risk of lung and pleural diseases, including non-malignant (e.g., pleural effusion, pleural plaque, rounded atelectasis, and asbestosis) and malignant (lung cancer and mesothelioma) diseases [40,41,42]. Occupational asbestos exposure is associated with a five-fold increase in the risk of lung cancer [43,44], and the risk of lung cancer has been noted to increase with increased exposure to asbestos [44]. Malignant mesothelioma (mainly malignant pleural mesothelioma) is the most common malignant occupational respiratory disease, and 23,000 cases of malignant mesothelioma were caused by asbestos exposure in 2015 [45]. Although all forms of asbestos fibers can also cause pulmonary fibrosis [39], which is referred to as asbestosis, the risk of asbestosis is higher following exposure to the amphibole group than to the serpentine group [46]. Asbestosis is a progressive disease and its severity is related to the total dose of asbestos exposure [46]. A recent study revealed that longer fibers are more powerful than shorter fibers in stimulating the nuclear factor (NF)-kB pathway and gene promoter activities, and the length of fibers is directly related to pathogenesis, with longer fibers generally being more harmful [47]. The fiber type at the time of initial exposure and physical properties of various asbestos fibers determine the toxic effects of asbestos [41,48,49,50]. Alveolar epithelial cells and alveolar macrophages internalize inhaled asbestos fibers, and consequently, oxidative stress is induced by these cells in the lung due to the release of reactive oxygen species (ROS) and reactive nitrogen species (RNS) [48,50,51]. Although the use of asbestos has been banned or strictly restricted in at least 40 countries [52], the health risks of asbestos remain relevant because the latent period between exposure and the development of asbestos-related diseases can span several decades [53]. 

Similar to asbestos, crystalline silicon dioxide or silica also induces pulmonary fibrosis, which is referred to as silicosis. People who work in the mining, quarrying, drilling, foundries, ceramics, and sandblasting industries are at high risk of developing silicosis [54]. Silicosis generally has a latent period of approximately 10–30 years; however, accelerated silicosis can develop earlier in people exposed to large quantities of fine silica dust during a relatively short period (typically months) [54,55,56,57,58]. Accelerated silicosis, also known as silicoproteinosis, may result from the excessive production of proteinaceous materials and surfactant proteins by hypertrophic type II pneumocytes and excessive radicals by silica particles [59,60,61,62]. Similar to asbestosis, it has been reported that chronic silicosis is also related to the generation of ROS and RNS due to silica particles lodging in alveolar macrophages [63]. 

Some occupational pollutants, such as sensitizers or irritants, cause or exacerbate WRA (Table 3). Asthma has the second highest prevalence and mortality rate after COPD among chronic respiratory diseases worldwide [64]. The incidence of asthma was 43 million in 2017, and the estimated prevalence and mortality were 273 million (3.6% of the world population) and 500,000 deaths (mortality rate of 6.48/100,000 population) [64,65,66], and approximately 7% of these 500,000 deaths were attributed to WRA [67]. WRA can be classified as occupational asthma (OA) or work-exacerbated asthma (WEA) [68,69]. OA is defined as asthma caused by exposure to certain materials in the occupational environment, whereas WEA is defined as preexisting or concurrent asthma exacerbated by exposure at work [68,69,70], and the prevalence of OA and WEA has been estimated at 16.0% and 21.5%, respectively [71]. In a surveillance study conducted in the United States, ‘miscellaneous chemicals and materials (including pesticides and glues)’, ‘mineral and inorganic dusts (including cement dust and copier toner) and ‘cleaning materials’ were the top three agent categories attributable to WEA, which accounted for >40% of all 14 categories of the Association of Occupational and Environmental Clinics Exposure Code List [71,72]. OA is further categorized into sensitizer-induced OA and irritant-induced OA [69]. Although sensitizer-induced OA accounts for approximately 90% of all OA cases and 5–18% of OA cases are reported as irritant-induced asthma [73], there are variations across studies. Approximately 600 agents have been associated with OA, of which 400 cause asthma due to sensitization mechanisms [74]. In addition, >70% of cases of sensitizer-induced OA are attributable to only eight agents (flour, latex, isocyanates, persulfates, metals, quaternary ammonium compounds, acrylates, and wood) [75]. The onset of symptoms after a latency period (e.g., months or years) is characteristic of sensitizer-induced OA, and exposure at higher levels, sensitizing agents, and individual characteristics are related to an earlier onset of OA [76]. Sensitizers include high molecular weight (>5 kDa) and low molecular weight (≤4 kDa) agents [77,78]. The former are mostly proteins, such as those of animal or plant origin, and have an IgE-mediated immune mechanism, while the latter are composed of organic or inorganic compounds that can act as haptens but often have an unclear mechanism [69,70]. The phenotype of sensitizer-induced asthma has been revealed in recent investigations, including a large cohort study [75,79]. In the group of patients exposed to low molecular weight sensitizers, symptoms such as chest tightness and daily sputum at work, severe asthma exacerbations, and late asthmatic reactions during specific inhalation challenge (SIC), were more frequent [75,79]. In the group of patients exposed to high molecular weight sensitizers, there were more cases of work-related conjunctivitis and rhinitis, baseline airway obstruction, atopy, early asthmatic reactions to SIC, higher levels of fractional exhaled NO, and blood eosinophils [75,79]. Irritant-induced asthma (also known as nonimmunologic asthma), including reactive airway dysfunction syndrome (RADS), occurs without a latent period after high-level single or multiple exposures to an irritant [68,80]. RADS was first reported in 1985 [81] and best meets the definitive diagnosis of irritant-induced asthma, i.e., onset within 24 h of exposure to high concentration irritants such as gases, fumes, smoke, or vapors, with pulmonary functional changes (positive bronchodilator response or methacholine challenge) [80,81,82,83]. Additionally, there is another phenotype of RADS, which is characterized by a more insidious onset >24 h after exposure due to exposure to multiple agents or chronic exposure at low concentrations [82,84,85]. The reported causative agents of irritant-induced asthma include benzene-1,2,4-tricarboxylic acid, 1,2-anhydride (trimellitic anhydride), chlorine, cobalt, cement, environmental tobacco smoke, grain, welding fumes, construction work, swine or poultry confinement or farming, and the 2001 New York World Trade Center collapse [86]. Neurogenic inflammation and oxidative stress are suggested mechanisms of irritant-induced asthma [82], for example, activated transient receptor potential channels [87], or irritant agents that enhance the airway response to allergens [88,89]. In addition, genetic factors, such as polymorphisms of genes related to inflammation via the NF-κB pathway, may also play a role in irritant-induced asthma [90]. 

In recent decades, it has been demonstrated that cleaning agents may cause asthma. The cleaning products frequently used around the world contain well-known sensitizers (e.g., quaternary ammonium compounds, amines, and fragrances) and irritants (e.g., sodium hypochlorite, hydrochloric acid, and alkaline agents (ammonia and caustic soda)). Therefore, these cleaning agents can be attributed to sensitizer-induced and irritant-induced asthma, in addition to WEA [91,92]. 

Certain organic chemicals and low molecular weight compounds can cause HP [93]. The incidence of HP ranges from to 1–3 to 30 per 100,000 people per annum [94,95,96]. It is estimated that HP due to occupational exposure accounts for 12-28% of all HP cases [5]. Most cases of HP are reported in farmers (1.3–12.9% [97,98]), bird breeders (3.7–10.4% [99,100]), and mushroom producers (3.5–29% [101,102,103,104]). Isocyanates are low-molecular-weight organic chemicals that are well-known HP-causing agents [105], and are often used as surface protective materials, in paints, and in polyurethane production [106]. Although isocyanates are not immunogenic per se, haptens consisting of human proteins may induce HP [105]. Although the immunopathogenesis involved in HP have not been fully elucidated, it is suggested that aberrant immunological mechanisms are involved. Inhaled antigens interact with antigen-presenting cells (APCs), such as macrophages and dendritic cells via pattern recognition receptors, leading to stimulation of the Th1 response of APCs. The Th1 response of APCs is enhanced by cytokine and chemokine production, which promotes the influx of neutrophils into the lungs; subsequently, the complement cascade is initiated and macrophages are stimulated, forming granulomas as a result of fusion with multinucleated giant cells and epithelioid cells. Concurrently, antibodies, particularly immunoglobulin G (IgG), are produced by stimulated B cells that recognize antigens and form antigen–antibody complexes. During the transition from acute, predominantly inflammatory HP to chronic or fibrous HP, fibroblast proliferation is promoted by chemotactic factors produced by granulomas, switching from a Th1 to Th2 inflammatory response, a decrease in regulatory T cells, production of cytotoxic CD8^+^ T cells, and differentiation of Th17 cells. Fibroblasts then differentiate into myofibroblasts, producing collagen and extracellular matrix components, which promotes pulmonary fibrosis [93]. In addition, gene variants, including the tumor necrosis factor (TNF)-gene promotor region [107], overexpression of mucin 5 subtype B (MUC5B) [108], and telomere dysfunctions [109], are involved as genetic risk factors in individuals susceptible to fibrotic HP. 

In recent years, new toxic materials have emerged, such as cross-linked acrylic acid-based polymers (CL-PAAs), which cause serious lung disorders, including pneumoconiosis. CL-PAAs are polymers that contain acrylic acid as a monomer. CL-PAAs are widely used as intermediates in the manufacture of pharmaceuticals and cosmetics [106]. According to reports from the Working Group on Occupational Accident Diseases of the Ministry of Health, Labor and Welfare in Japan, all affected workers with lung diseases were relatively young (aged 20–40 years), and pneumoconiosis developed within a short period (several years) from the initial CL-PAA exposure before progressing. The radiological and clinical findings in these workers were pulmonary fibrosis, interstitial pneumonia, emphysema, and pneumothorax [110]. Since CL-PAAs have not been reported to be harmful to the lung, they should be considered a new hazard and further investigations should be conducted to determine the mechanism of pulmonary toxicity following CL-PAA exposure.

**Table 2 ijerph-19-02788-t002:** Representative occupational pollutants with health effects on the respiratory system (except work-related asthma) in this review.

Pollutants	Occupational Exposure Circumstances	Respiratory Diseases	Reference Number
Inorganic substances			
Mineral dusts			
Asbestos Chrysotile (Serpentine group) Crocidolite, amosite, tremolite, and others (Amphibole group)	Construction, manufacturing brake lining and pads, and handling insulation	Lung cancer, mesothelioma, pleural effusion, pleural plaque, rounded atelectasis, and asbestosis	[38,39,40,41,42]
Crystalline silicon dioxide, Crystalline silica	Mining, quarrying, drilling, foundries, ceramics, and sandblasting industries	Silicosis, Lung cancer	[54,55,56,57,58]
Heavy metals			
Arsenic	Hot smelting, fur handling, manufacturing sheep-dip compounds and pesticides, and vineyard working	Lung cancer	[7,38]
Chromium [VI]	Producing chromate, chromium platers and ferrochromium, and manufacturing chromate pigment	Lung cancer	[7,38]
Nickel	Mining, smelting, and electrolyzing	Lung cancer	[7,38]
Radiation			
Low-LET radiation; X-rays, and γ-rays	Medical professions and nuclear industry	Lung cancer	[7,38]
High-LET radiation and Radon	Underground mining	Lung cancer	[7,38]
Organic substances			
*Saccharomycetes* spp., *Aspergillus* spp.	Farming	Hypersensitivity pneumonitis	[93]
Bird bloom, feather, droppings, serum, and down products	Bird breeding, Manufacturing down products	Hypersensitivity pneumonitis	[93]
Shitake, bunashimeji, nameko, eringi, and thermophilic *Actinomycetes*	Mushroom working	Hypersensitivity pneumonitis	[93,111]
Isocyanates	Surface protective materials, painting, and producing polyurethane	Hypersensitivity pneumonitis	[93,106]
Cross-linked acrylic acid-based polymers	Manufacturing pharmaceuticals and cosmetics	Pneumoconiosis, emphysema, and pneumothorax	[106,110]

LET: liner energy transfer.

**Table 3 ijerph-19-02788-t003:** Work-related asthma (WRA) and causative agents.

Classification of WRA	Causative Agents	Reference Number
Work-exacerbated asthma (WEA)	Miscellaneous chemicals and materials (including pesticides and glues), mineral and inorganic dusts (including cement dust and copier toner), and cleaning materials	[71,72,91,92]
Occupational asthma (OA)		
Sensitizer-induced OA	High molecular weight (≧5 kDa); flour, latex	[75,91,92]
	Low molecular weight (<5 kDa); isocyanates, persulfates, metals, cleaning materials (quaternary ammonium compounds, amines, and fragrances), acrylates, and wood	
Irritant-induced OA (including RADS)	Benzene-1,2,4-tricarboxylic acid, 1,2-anhydride (trimellitic anhydride), sodium hypochlorite, hydrochloric acid, alkaline agents (ammonia and caustic soda), chlorine, cobalt, cement, environmental tobacco smoke, grain, welding fumes, construction work, swine or poultry confinement or farming, and the 2001 New York World Trade Center collapse	[68,80,86,91,92]

## 4. Conclusions

Respiratory diseases related to ambient or occupational pollution can be avoided, and it is crucial to avoid exposure to all harmful materials in order to prevent and control such diseases. Adequate awareness of the risk of exposure to hazardous materials should be mandatory for all employers, occupational and pulmonary physicians, and researchers. Adequate education on occupational health and safety and awareness of environmental quality may lead to better health promotion and behavioral changes to prevent related diseases and disease exacerbations, as well as have an impact on social behavior to reduce air pollutants.

## Data Availability

Data sharing not applicable. No new data were created or analyzed in this study. Data sharing is not applicable to this article.

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
