# Peer review of "The Impact of Ambient Environmental and Occupational Pollution on Respiratory Diseases"

_ijerph, 2022, doi:10.3390/ijerph19052788_

Round 1
Reviewer 1 Report
The manuscript entitled "The impact of ambient environmental and occupational pollution on respiratory diseases" describes lung and respiratory diseases in relation to ambient pollutants, particularly PM2.5, and occupational air pollutants, excluding communicable diseases and indoor pollutants, including tobacco smoke exposure.
The author aims to increase the awareness of pollutants and reduce the development and exacerbation of diseases caused by air pollutants are paramount to addressing ambient and occupational pollution.
However, there are still some questions to be considered as follows:
Specific comments:
1. The introduction of the research background is not detailed enough. The author did not clearly point out the current research status and existing problems, nor did they explain why the review research was carried out.
2. Which bibliometric tool is selected in this study should be introduced in the method part.
3. The main research contents of the article are almost of the nature of popular science, and there are no substantive results and conclusions.
4. In addition, there are no charts and tables in the full text, which can not intuitively show the review results to readers.
Author Response
We have revised our manuscript according to the detailed advices from you.
Please see the attachment.

Reviewer 2 Report
Nowadays, air pollution is a common phenomenon and it is often correlated with chronic diseases. There are many studies and statistical research that demonstrate the direct correlation between chemical substances and diseases. In the current situation of increasing concentrations of air pollutants, international and national services were forced to introduce regulations, monitoring and controls. This enables specific values to be attributed to the human body.
In the manuscript submitted by Nishida and Yatera, describes lung and respiratory diseases in relation to ambient pollutants, particularly PM2.5, and occupational air pollutants, excluding communicable diseases and indoor pollutants, including tobacco smoke exposure.
A review of the current literature related to the topic of air pollution and health was used. The databases used included PubMed. Numerous studies have confirmed negative impact of air pollution on the health people.
In the literature on the subject, this problem has been discussed many times. In this respect, the work is not original. However, I find the approach to the problem interesting.
Comments and suggestions:
- The aim of the study is to review individual studies described in the literature. The importance of the presented results in the context of human health should be emphasized more.
- In conclusions, the authors wrote: ”Adequate awareness of the risk of exposure to hazardous materials should be mandatory for all employers, occupational and pulmonary physicians, and researchers”. Perhaps a sentence should be added how important is the awareness (education) of the whole society regarding the risk of exposure to hazardous materials.
- Technical notes: Line 32: (PM) 2.5 (PM 2,5); Lines 364 and 366: date (2020) – bold font.
Author Response

(The authors gave the same response as above.)

Round 2
Reviewer 1 Report
Accept in present form.